# The Impact of Digital Learning Competence on the Academic Achievement of Undergraduate Students

**DOI:** 10.3390/bs15070840

**Published:** 2025-06-22

**Authors:** Yafeng Song, Shuqi Lv, Meng Wang, Zhuoxi Wang, Wei Dong

**Affiliations:** School of Education, Tianjin University, Tianjin 300350, China; lvshuqi02_@tju.edu.cn (S.L.); wangmeng17@tju.edu.cn (M.W.); zhuoxiwang2000@gmail.com (Z.W.)

**Keywords:** undergraduate students, digital learning competence, academic achievement, impact study

## Abstract

Digital learning competence has gradually become one of the core qualities essential for undergraduate students. To effectively enhance undergraduates’ digital learning abilities and their positive impact on academic performance, this study developed a validated survey on digital learning competence and academic achievement. A total of 312 valid questionnaires were collected from undergraduate students. Descriptive statistical analysis revealed that the overall academic achievement of the sample students was at an upper-middle level, with course achievements and practical achievements being higher than scholarly achievements. Differential analysis showed that male students scored higher than female students in scholarly achievements, practical achievements, and overall academic performance. Additionally, senior students generally outperformed junior students in course achievement, academic research, and overall academic performance, while undergraduates from key universities generally achieved higher academic results than those from ordinary undergraduate institutions. Correlation and regression analyses indicated that digital learning evaluation competence as a sub-competence under digital learning competence has significant positive predictive effects on undergraduates’ academic achievement. When other factors remained constant, for each unit increase in digital learning evaluation ability, academic achievement increased by 0.480 units. Therefore, universities can improve existing student development processes through measures such as enriching carriers, optimizing methods, and creating supportive environments to foster undergraduates’ digital learning competence, thereby enhancing their academic achievement.

## 1. Introduction

Computer information technology has seen widespread application in education, profoundly transforming educational models, teaching methods, and learning approaches. Digital technology addresses time and space limitations, giving rise to remote education ([46]), online education ([64]), and blended learning ([14]; [74]). Digital tools such as ChatGPT4.0 and Bing Chat ([36]), virtual reality ([19]; [30]), and augmented reality ([18]) have expanded students’ learning channels. Traditional teacher-centered classroom instruction is gradually being replaced by student-centered interactive, participatory teaching ([42]; [75]). Artificial intelligence applications such as intelligent tutoring systems, educational robots, and personalized learning recommendation systems greatly enhance student autonomy and initiative ([9]). AI tools, including online intelligent systems, collaborative robots, and chatbots, also improve teaching efficiency and quality ([8]).

The COVID-19 pandemic forced universities to shift away from traditional face-to-face teaching models ([55]), accelerating the popularization of online education. Educational digitalization is gradually becoming normalized. The European Union officially issued the Digital Education Action Plan (2021–2027) in 2020, proposing that high-quality, accessible digital education be widely recognized throughout Europe and supporting member states’ education and training systems to adapt to the digital age ([13]). In 2021, the Organisation for Economic Co-operation and Development (OECD), using Hungary’s higher education digital transformation as a pilot, published Supporting the Digital Transformation of Higher Education in Hungary, elevating higher education digitalization to a national strategy and constructing policy frameworks and action pathways ([47]). The United Nations Education Transformation Summit in September 2022 released the Action Initiative to Ensure and Improve the Quality of Universal Public Digital Learning, emphasizing that the international community needs to recognize current trends in higher education digital transformation, and calling for countries to establish new international rules, leverage digital technology advantages to empower teaching and learning, and create digital learning platforms ([67]).

As the primary subjects of educational activities, students’ digital competence has gradually gained widespread attention from governments, society, and academia. The EU’s DigComp framework is currently recognized as a digital competence framework in academia, and is widely used in relevant research ([44]; [6]). The latest DigComp 2.2 defines digital competence as “the confident, critical, and responsible use of and engagement with digital technologies for learning, working, and participating in society”. It includes five dimensions—information and data literacy, communication and collaboration, digital content creation, safety, and problem-solving. Digital competence is one of the core competencies for 21st-century citizens and a key interdisciplinary skill for lifelong learning, employment, and active social participation ([68]). Numerous studies demonstrate that digital competence positively impacts student development, for example, by increasing the acceptance of digital learning ([56]), promoting learning behaviors ([50]), enhancing digital informal learning and academic performance ([40]), improving English learning outcomes ([45]), increasing online learning satisfaction and thereby improving academic achievement ([77]), enhancing physical education performance ([33]), and improving undergraduates’ research capabilities ([41]).

However, existing research primarily focuses on evaluating students’ overall digital competence ([15]; [71]; [65]; [39]). Limited attention has been paid to students’ Digital Learning Competence (DLC) in specific learning contexts. Thus, the mechanism by which digital learning competence influences academic achievement remains insufficiently explored. Undergraduates are often involved in digital competence research ([79]). Therefore, this paper focused on undergraduate students and developed a valid and reliable digital learning competence assessment scale based on a comprehensive literature review. Through questionnaire surveys and statistical analysis, this study aims to address the following questions:What is the structural connotation of undergraduates’ digital learning competence?Which dimensions of digital learning competence have a significant impact on undergraduates’ academic achievement?How can we improve undergraduates’ digital learning competence to enhance their learning outcomes and academic performance, thereby improving talent cultivation quality?

## 2. Literature Review

### 2.1. Digital Learning Competence

The term “E-learning” gained popularity in the late 1990s. In 2000, the CEO Forum on Educational Technology (ET-CEO Forum) defined digital learning as an approach that integrates digital technology with curriculum content, emphasizing the construction of digital learning environments, resources, and methodologies that match the talent requirements of the 21st century ([5]). The OECD’s PISA 2025 framework highlights that “learning in the digital world” not only requires students have the ability to use digital tools for knowledge construction and problem-solving, but it also emphasizes their self-regulation capabilities in metacognition, behavioral regulation, and emotional management ([73]). In research on digital learning competence, we need to clearly distinguish related concepts such as “digital literacy”, “digital competence”, and “digital maturity”. Digital literacy originated from the needs of the information society. It refers to the ability to access, evaluate, generate, and communicate information using digital technologies in effective and efficient ways ([17]). It encompasses various abilities, including computer literacy, information and communication technology literacy, information literacy, and media literacy ([34]). Digital literacy represents a fundamental cognitive ability. Digital maturity emphasizes the overall development level of organizations or individuals in digital technology application processes. It involves deep-level aspects such as strategic planning, cultural adaptation, and risk mitigation ([53]; [28]). In contrast, digital competence is a more comprehensive concept that covers individuals’ abilities to complete various tasks in digital environments, and includes five dimensions—information and data literacy, communication and collaboration, digital content creation, safety, and problem-solving ([68]).

Digital learning competence focuses on specific learning contexts. Research has defined digital learning competence as learners’ ability to conduct effective learning using digital tools in digital environments. It represents a combination of knowledge, skills, and attitudes ([76]; [27]). Since digital learning competence is often developed by integrating virtual learning environments, digital learning tools, and methods ([3]), this study defined digital learning competence as the ability to conduct learning activities in digital learning environments, such us utilizing digital learning resources and digital learning tools for knowledge acquisition, construction, management, and evaluation, thereby achieving improvements in cognition, knowledge, skills, attitudes, and other aspects.

### 2.2. Academic Achievement

Academic achievement is a crucial indicator for measuring students’ learning performance and educational outcomes. In a narrow sense, academic achievement often equates with students’ learning grades ([10]; [78]). In a broader sense, the definition encompasses students’ comprehensive performance throughout the educational process, including interpersonal relationships ([37]) and career planning abilities ([72]) among the measurement indicators. With societal development and the urgent demand for high-quality talent, higher education goals are becoming increasingly diverse and comprehensive. The OECD emphasizes that future education should focus on cultivating individuals with key competencies such as collaboration, creativity, self-regulation, and value judgment ([48]). UNESCO points out that higher education should be committed to promoting students’ comprehensive development, covering multiple dimensions including academic ability, social responsibility, sustainable development awareness, and lifelong learning ability ([66]).

Therefore, this paper defined academic achievement as the comprehensive outcome students obtain through learning activities during a certain period. It encompasses learning results, learning behaviors, and learning attitudes, and covers three aspects—course achievement, scholarly achievement, and practical achievement.

### 2.3. Relationship Between Digital Learning Competence and Academic Achievement

Academic achievement is often a key element in research. School climate, family background, teacher support, learning motivation, and self-regulation abilities ([11]; [16]; [62]; [31]; [7]; [21]) are common variables in current research. With the development of educational informatization, scholars have gradually focused on the impact of emerging media and technologies on academic achievement. Research indicates that online communication in online learning systems ([59]), environmental compatibility and personal innovation ([70]) significantly influence students’ academic performance. The more frequent use of digital textbooks in classrooms can improve students’ academic performance, academic interest, and learning skills ([32]). Compared to students who use personal computers individually at home, students without personal computers can significantly enhance both their current and their long-term academic performance by frequently using information and communication technology (ICT) with teachers in classrooms ([20]). In higher physical education, integrating gamified digital game-based learning (DGBL) with the ARCS model can effectively improve student motivation, academic performance, and practical knowledge ([4]). [23] ([23]) found through empirical research that digital capabilities not only directly affect students’ academic performance, but also indirectly influence it by enhancing students’ learning self-efficacy. [63] ([63]) discovered that undergraduate students’ media literacy positively impacts perceived learning outcomes, with critical production having the most significant effect on learning outcomes. [54] ([54]) found that health science students’ digital learning competence not only enhances their comprehensive abilities, but also broadens their understanding of related skills. [49] ([49]) analyzed PISA 2018 data, and found that students’ interaction behaviors with ICT significantly correlate with science academic achievement. Positive ICT attitudes and frequent educational ICT use have positive effects on improving science performance, while excessive recreational ICT use may produce negative impacts.

However, the current research on the relationship between digital learning competence and academic achievement still has limitations. Existing studies have not yet established universally accepted evaluation standards for digital learning competence and, lack further exploration of its specific impact on academic achievement. Therefore, this study focused on undergraduate students by investigating the influence of digital learning competence on academic achievement, aiming to provide new perspectives on the research in the relevant field.

## 3. Methodology

### 3.1. Research Design

We can collect data through questionnaires designed to study specific populations’ behaviors, attitudes, opinions, knowledge, or characteristics. This enables the rapid, systematic collection of large amounts of data for quantitative analysis. This study employed the questionnaire survey method, designing qualified questionnaires to collect data from Chinese undergraduate students, aiming to analyze the relationship between digital learning competence and academic achievement. Specifically, this study reviewed relevant research, summarized existing questionnaire measurement indicators, and independently developed valid and reliable questionnaires. It investigates undergraduates’ current digital learning competence and academic achievement status, explores the impact of undergraduates’ digital learning competence on academic achievement, draws research conclusions, and proposes corresponding countermeasures. The questionnaire targets undergraduate students and primarily consists of three parts—demographic information, a digital learning competence scale, and an academic achievement survey. The questionnaire survey is divided into the preliminary survey and formal survey phases. The preliminary survey distributes questionnaires on a small scale, conducts item analysis and reliability–validity analysis of the questionnaire data, and forms the final formal questionnaire after eliminating invalid items. The formal questionnaire survey is then conducted on a large scale. SPSS18 is used for the descriptive statistical analysis, differential analysis, correlation analysis, and regression analysis of questionnaire data. This helps us investigate undergraduates’ current digital learning competence and academic achievement status and explore the correlation between undergraduates’ digital learning competence and academic achievement, and provides valid references for the research’s conclusions.

### 3.2. Development of Measurement Tools

#### 3.2.1. Development of Digital Learning Competence Scale

Currently, in the relevant research, there is no unified standard for the dimensional classification of digital learning competence. The UK’s *Jisc Digital Capability Framework*, based on the EU digital competence framework, particularly emphasizes “learning-type” capability and “lifelong learning” concepts. It proposes six elements of digital capability for higher education and vocational education teachers and students, as follows: ICT/digital proficiency, information data and media literacy (critical use), digital creation, problem-solving and innovation (creative production), digital communication, collaboration and participation (participation), digital learning and development (development), and digital identity and wellbeing (self-realization) ([26]). The International Society for Technology in Education (ISTE) digital competence strategy emphasizes students’ identity recognition in digital learning. It uses various standards to measure students’ digital competence, depending on whether they can do the following: recognize responsibilities and opportunities to contribute to digital communities; use digital tools to rigorously manage various resources to build knowledge, produce creative works, and create meaningful learning experiences for themselves and others; use various technologies in the design process to identify and solve problems through creating new, useful, or imaginative solutions; leverage technological approaches to develop and test solutions, thereby formulating and applying strategies for understanding and solving problems; use platforms, tools, styles, formats, and digital media appropriate to their goals to communicate clearly and express themselves creatively for various purposes; use digital tools to broaden perspectives and enrich their learning by collaborating with others and working effectively in local and global teams ([24]). [76] ([76]) developed a digital learning competence framework that was effective in assessing secondary school students’ digital learning competence, including six dimensions—technology use, cognitive processing, digital reading skills, time management, peer management, and will management. [52] ([52]) developed and validated Digitest, a digital learning competence assessment scale specifically for primary and junior high school students, comprising nine dimensions—perceptual control, behavioral attitudes, behavioral intentions, creating digital materials or content, programming digital content, communication in the digital world, operating with digital tools, protecting oneself and others in the digital world, and legal behavior in the digital world. [61] ([61]) assessed the digital learning competence of secondary vocational school students via five dimensions—cognitive processing and reading, technology use, thinking ability, activity management, and will management.

Although the aforementioned frameworks provide multi-dimensional perspectives on the composition of digital learning competence, several limitations remain. First, some current models focus primarily on basic education stages, which are insufficient for undergraduate stages wherein learning environments and tasks are relatively complex. There is still a lack of digital learning competence frameworks suitable for undergraduate stages that encompass the higher-order abilities of university students in academic research, knowledge construction, and digital expression. Second, the dimensional divisions in some frameworks are unclear. For example, Yang et al. separately treated three closely related management dimensions—time management, peer management, and volition management. Pedaste et al. introduced non-core learning dimensions such as “protecting oneself and others in the digital world” and “legal behavior in the digital world” into their framework, which may reduce the targeting of digital learning competence assessment. Finally, the theoretical foundations and dynamic nature of current models require strengthening. Some models, such as the Jisc digital competence framework and ISTE Digital Competence strategies, are built on policy advocacy or technical standards, while lacking the integration of core learning science theories, such as Self-Regulated Learning Theory and Cognitive Load Theory, making it difficult to reveal the mechanisms between different competence dimensions in digital environments. Meanwhile, the emergence of advanced technologies, such as generative AI and large language models, also poses challenges to existing digital learning competence models.

Self-Regulated Learning (SRL) is a core conceptual framework for understanding cognitive, motivational, and emotional aspects of learning. It has become one of the most important research areas in educational psychology ([51]). Therefore, this study critically integrated the above digital learning competence frameworks by combining them with [80]’s ([80]) Self-Regulated Learning Theory (SRL). We proposed a digital learning competence framework containing five dimensions—awareness, technical skills, behaviors, management, and evaluation (Table 1). [80]’s ([80]) SRL model divides the learning process into three phases: Forethought, Performance, and Self-reflection. In the Forethought phase, students analyze learning tasks, clarify learning goals, and formulate corresponding learning plans. Students’ motivational beliefs (such as confidence and task value) play key roles in this phase. They not only provide the psychological driving force for learning activities, but they also influence the selection and activation of subsequent learning strategies. In the Performance phase, students begin specific task execution while monitoring learning progress and adopting various self-control strategies to maintain cognitive engagement and task motivation. In the Self-reflection phase, students evaluate their task completion and attribute causes of success or failure. Positive attribution can have positive effects on future learning attitudes and behaviors, while negative attribution can have negative effects on future learning attitudes and behaviors.

In the five-dimensional framework proposed by this study, digital learning awareness corresponds to the Forethought phase in SRL. It reflects individuals’ cognition, understanding, and attention to digital learning, as well as attitudes toward digital learning tools and resources. It also directly determines whether individuals are willing and able to actively participate in digital learning, serving as the prerequisite for stimulating subsequent behaviors. Digital learning technical skills, digital learning engagement behaviors, and digital learning self-management collectively correspond to the Performance phase. Technical skills represent individuals’ skills and abilities in using digital technologies and tools, serving as the foundation for achieving digital learning. Engagement behaviors represent the series of behaviors and habits individuals display during digital learning processes, including collaboration, communication, and connection abilities, which can reflect individuals’ digital learning states. Self-management represents individuals’ abilities to manage learning resources, learning tools, learning time, and learning emotions during digital learning processes, helping individuals control the quality of digital learning and promote efficient learning. Digital learning evaluation competence highly aligns with the Self-reflection phase. It refers to individuals’ abilities to reasonably and accurately evaluate digital learning resources, tools, environments, processes, and outcomes, helping individuals reflect on and optimize their learning methods and processes to achieve continuous improvement. These five dimensions are interrelated and completely cover the “Forethought–Performance–Self-reflection” cycle of SRL theory. They represent a systematic expression of learners’ self-regulation abilities in digital learning contexts, suitable for measuring and enhancing university students’ digital learning competence in higher education.

The undergraduate digital learning competence scale is shown in Table 1. The scale contains 31 items using a 7-point Likert scale. Strongly agree scores 7 points, somewhat agree scores 6 points, slightly agree scores 5 points, uncertain scores 4 points, slightly disagree scores 3 points, somewhat disagree scores 2 points, and strongly disagree scores 1 point. Based on the initial questionnaire development, first, the critical ratio method was adopted. The total scores of scale items were calculated and ranked. The top 27% of scores were designated as the high-score group, and the bottom 27% as the low-score group. An independent samples t-test was then used to compare differences between high- and low-scoring groups. If *p* < 0.05, the high- and low-scoring groups showed significant differences. If differences were not significant, the item could not discriminate respondents’ response levels, indicating the item should be modified or deleted. The results indicate that all items except B4 showed significance (*p* < 0.05). Therefore, item B4 was deleted while other items were retained. Second, item–total correlation analysis was conducted using Pearson correlation coefficients to measure correlations between each item and total scores. High correlations indicate high consistency between items and the overall scale. Items with Pearson correlation coefficients below 0.3 can be adjusted or deleted. All items in this scale’s item–total correlation analysis had correlation coefficients exceeding 0.3. Therefore, all items were retained. Third, validity analysis was performed. The Kaiser–Meyer–Olkin (KMO) value of the digital learning competence scale was 0.711, indicating scale items were suitable for factor analysis. Principal component analysis and varimax rotation were used for exploratory factor analysis. After removing six items (A7, C1, C5, C6, D5, E6) that did not meet dimensional expectations, the scale was divided into five factors. Factor one includes A1, A2, A3, A4, A5 and A6; factor two includes B1, B2, B3, B5, B6 and B7; factor three includes C2, C3 and C4; factor four includes D1, D2, D3 and D4; factor five includes E1, E2, E3, E4 and E5, totaling 24 items. Finally, reliability analysis was conducted. The overall reliability and dimensional reliability test results of the final digital learning competence scale show the following: digital learning competence Cronbach’s Alpha = 0.890, factor one digital learning awareness Cronbach’s Alpha = 0.864, factor two digital learning technical skills Cronbach’s Alpha = 0.817, factor three digital learning engagement behaviors Cronbach’s Alpha = 0.730, factor four digital learning self-management Cronbach’s Alpha = 0.788, and factor five digital learning evaluation competence Cronbach’s Alpha = 0.778. Overall, the scale demonstrates good reliability and can be used to measure undergraduate students’ digital learning competence.

#### 3.2.2. Development of the Academic Achievement Questionnaire

This study defines academic achievement via comprehensive outcomes encompassing learning results, learning behaviors, and learning attitudes that students obtain through learning activities within a certain period. It covers three aspects: course achievement, scholarly achievement, and practical achievement. Course achievement refers to outcomes students achieve in professional course learning, reflecting the mastery of disciplinary foundational knowledge and skills. It emphasizes systematicity and the standardization of the learning process. The measurement indicators include professional ranking and College English Test Band 4 and Band 6 scores. Scholarly achievement refers to outcomes from students’ participation in research activities, academic exchanges, and disciplinary competitions. It focuses on higher-order cognitive abilities and innovative practice, reflecting academic literacy and problem-solving capabilities. Measurement indicators include participation in research projects, academic conferences, and disciplinary competitions with awards. Practical achievement refers to outcomes formed through students’ social practice, volunteer service, and professional experience activities. It emphasizes knowledge application and social responsibility, reflecting extensions of learning behaviors and the externalization of attitudes. Measurement indicators include participation in social practice, volunteer service, holding positions, participating in off-campus internships (not organized by schools), participating in sports competitions with awards, and participating in arts competitions with awards. The questionnaire adopts the single-choice format with 12 items in total.

### 3.3. Data Collection

The formal questionnaire consists of three parts: personal basic information, the undergraduate digital learning competence scale, and the undergraduate academic achievement questionnaire (Appendix A). The personal basic information section contains 4 demographic items: gender, grade, disciplinary category of major, and school level. The undergraduate digital learning competence scale is divided into 5 dimensions: digital learning awareness, digital learning technical skills, digital learning engagement behaviors, digital learning self-management, and digital learning evaluation competence, totaling 24 items. The undergraduate academic achievement questionnaire is divided into three parts: course achievement, scholarly achievement, and practical achievement, totaling 12 items.

The formal questionnaire distribution primarily employed online student self-assessment. The digital learning competence scale comprised a 7-point Likert scale; strongly agree scores 7 points, somewhat agree scores 6 points, slightly agree scores 5 points, uncertain scores 4 points, slightly disagree scores 3 points, somewhat disagree scores 2 points, and strongly disagree scores 1 point. Students’ final digital learning competence scores are the sum of scores from 24 items. Scores for digital learning awareness, digital learning technical skills, digital learning engagement behaviors, digital learning self-management, and digital learning evaluation competence are respectively the sum of scores from items in each dimension. The academic achievement questionnaire adopts the single-choice format. Each item is scored from low to high according to the selected answers (for example, professional ranking below 80% scores 1 point, 60–80% scores 2 points, 40–60% scores 3 points, 20–40% scores 4 points, top 20% scores 5 points). Course achievement scores are the sum of 3 items in that section. Scholarly achievement scores are the sum of 3 items in that section. Practical achievement scores are the sum of 3 items in that section.

Undergraduate education strives for comprehensive development in moral, intellectual, physical, aesthetic, and labor aspects. Using the case institution’s undergraduate comprehensive quality assessment plan as an example, moral education accounts for 8%, intellectual education accounts for 80%, and physical education, aesthetic education, and labor education each account for 4%. Academic achievement is operationally defined as: Total academic achievement score = Total course achievement score × 50% + Total Scholarly Achievement score × 30% + Total practical achievement score × 20%. This calculation method highlights the core position of course foundations while emphasizing the coordinated development of academic innovation and practical application. It aligns with the OECD’s educational goal of “cultivating collaboration, innovation, and self-regulation abilities” ([48]), and is consistent with UNESCO’s concept of “promoting students’ comprehensive development” ([66]).

The formal distribution of the questionnaire was primarily conducted online, with a total of 350 responses collected. Of these, 312 were valid. Regarding the gender composition of the sample, there were 133 males, accounting for 43% of the total sample, and 179 females, accounting for 57%, with a difference of 46 individuals between the two groups. In terms of academic year, the largest group was from the fourth year, comprising 59% of the total sample. The second and third-year students accounted for 14% and 15%, respectively, while fifth-year students comprised the smallest group, at 3%. In terms of academic discipline, the largest group was from the field of management, comprising 22% of the sample, followed by education students at 17%. Students from the fields of science and engineering were represented in similar numbers, while those from the military and the agricultural sciences were the smallest groups. Finally, regarding the institutional level of the schools represented, the distribution was relatively even, with students from Double First-Class universities and other universities comprising nearly equal proportions of the sample.

## 4. Results

Statistical analysis is a process of collecting, processing, analyzing, and interpreting data using statistical principles and methods. It can scientifically process large amounts of data and extract valuable information from them. This article mainly uses SPSS to conduct descriptive statistical analysis, difference analysis, correlation analysis, and regression analysis on the questionnaire survey data to investigate the current status of undergraduate students’ digital learning ability and academic achievement, and explore the correlation between undergraduate students’ digital learning ability and academic achievement.

### 4.1. Descriptive Statistical Analysis

Table 2 presents a descriptive statistical analysis of undergraduate digital learning competence and academic achievement across various dimensions. The overall mean score for digital learning competence was 5.54 (SD = 0.47), indicating that undergraduates demonstrate relatively high levels of digital learning competence with minimal internal sample variation and an overall concentration of scores. Specifically, digital learning engagement behaviors (M = 5.85, SD = 0.66) and digital learning awareness (M = 5.80, SD = 0.66) received the highest scores with smaller standard deviations, indicating students performed best and most consistently in goal setting, process monitoring, and technology acceptance. Digital learning technical competence had the lowest mean and highest standard deviation (M = 5.06, SD = 0.76), suggesting students performed worst in technology utilization with significant individual differences. Additionally, digital learning self-management (M = 5.57, SD = 0.65) and evaluation competence (M = 5.59, SD = 0.71) both scored above the overall average, indicating that students performed at upper-middle levels in digital learning resource integration and outcome reflection.

The overall mean for academic achievement was 8.22 (SD = 2.06), indicating that undergraduate academic achievement was at an upper-middle level, though development imbalances existed across different dimensions. Practical achievement had the highest mean and greatest dispersion (M = 15.78, SD = 4.44), demonstrating significant student accomplishments in practical activities but with notable individual differences. Course achievement had a mean of 9.32 (SD = 2.62), indicating good student performance in coursework. Academic achievement had the lowest mean of only 5.49 (SD = 2.39), revealing substantial room for improvement in research output and academic competitions.

### 4.2. Differential Analysis

To explore the mechanism by which digital learning competence affects academic achievement, this study conducted differential analyses across multiple dimensions of the survey data. The specific results are as follows (Table 3).

First, regarding gender differences, independent sample t-tests revealed no significant difference in course achievement between genders (*p* > 0.05). However, significant differences were found in academic achievement, practical achievement, and overall academic achievement (*p* < 0.05). These differences may stem from gender-based variations in intelligence types, learning motivation, social expectations, and educational approaches.

Second, regarding grade-level differences, one-way ANOVA (Analysis of Variance) revealed no significant difference in practical achievement across grade levels (*p* > 0.05). However, significant differences were found in course achievement, academic achievement, and overall academic achievement (*p* < 0.05). This may be attributed to senior students accumulating more scientific knowledge, learning methods, learning techniques, and learning resources during their extended digital learning experiences, which contributes to improved academic achievement levels.

Third, regarding disciplinary differences, one-way ANOVA revealed no significant differences in course achievement, academic achievement, practical achievement, or overall academic achievement across different disciplines (*p* > 0.05). This indicates that the influence of digital learning competence on academic achievement has cross-disciplinary universality.

Finally, regarding university tier differences, one-way ANOVA revealed significant differences in course achievement, academic achievement, practical achievement, and overall academic achievement across different university tiers (*p* < 0.05). This may be due to key universities having advantages in terms of digital teaching resources, faculty strength, and learning environments. Additionally, differences in student quality and learning potential exist among different universities.

### 4.3. Correlation and Regression Analysis

#### 4.3.1. Correlation Analysis

Correlation analysis can quantify the strength and direction of linear relationships between variables. Through correlation analysis, this study found the following results (Table 4).

Course achievement showed a significant positive correlation with digital learning evaluation competence, demonstrating significance at the 0.05 level. However, it showed no correlation with digital learning awareness, technical competence, behavioral competence, management competence, or overall digital learning competence.

Academic achievement demonstrated significant positive correlations with digital learning technical competence, management competence, evaluation competence, and overall digital learning competence. The correlations with digital learning technical competence, management competence, and evaluation competence all showed significance at the 0.01 level, while the correlation with overall digital learning competence showed significance at the 0.05 level. Academic achievement showed no correlation with digital learning awareness or behavioral competence.

Practical achievement exhibited significant positive correlations with digital learning technical competence, evaluation competence, and overall digital learning competence, all with significance at the 0.01 level. No correlations were found between practical achievement and digital learning awareness, behavioral competence, or management competence.

Overall academic achievement demonstrated significant positive correlations with digital learning technical competence, management competence, evaluation competence, and overall digital learning competence, all with significance at the 0.01 level. No correlations were found between overall academic achievement and digital learning awareness or behavioral competence. 

#### 4.3.2. Regression Analysis

As shown in Table 4, overall academic achievement demonstrated significant positive correlations with three dimensions of digital learning competence: technical competence, management competence, and evaluation competence. To investigate the quantitative relationships between academic achievement and these three dimensions, this study used academic achievement as the dependent variable and digital learning technical competence, management competence, and evaluation competence as independent variables. Gender, grade level, and university tier—demographic variables that showed significant differences in academic achievement—were used as control variables. Stepwise multiple linear regression analysis was employed, with the results shown in Table 5.

Table 5 indicates that the effects of digital learning technical competence and management competence on academic achievement were not significant; therefore, they were removed and did not enter the regression equation. The model passed the F-test (F = 17.462, *p* = 0.000). After controlling for the confounding effects of gender, grade level, and university tier, based on the multiple regression model Y = β_0_ + β_1_X_1_ + β_3_X_3_ + … + β_k_X_k_ + β_m+1_Z_1_ + β_m+2_Z_2_ + … + β_k+m_Z_m_ + ε, substituting the results from the table yieldsY = 4.673 + 0.480X_1_ + 1.458Z_1_ − 0.638Z_2_ + 0.510Z_3_
where Y represents academic achievement, X_1_ represents digital learning evaluation competence, Z_1_ represents “Double First-Class” universities, Z_2_ represents lower grade levels, and Z_3_ represents male gender. The regression equation indicates that digital learning evaluation competence has a positive impact on undergraduate academic achievement. When other factors remain constant, for each standard unit increase in digital learning evaluation competence, academic achievement increases by 0.480 standard units.

## 5. Discussion

This study focused on undergraduate students, examining the multidimensional construction of digital learning competence and its impact on academic achievement. By systematically integrating core elements, including information technology skills, learning strategies, and self-regulation, it innovatively proposes a five-dimensional digital learning competence framework encompassing awareness, technology, behavior, management, and evaluation. This framework not only enriches structural model research on digital learning competence, but also provides a scientific and feasible tool for assessing digital learning competence in higher education. It fills a research gap in this field at the level of higher education, offering significant theoretical contributions and practical application value.

Regarding theoretical contributions, firstly, this research has expanded the theoretical boundaries of digital learning competence. Traditional digital learning competence research primarily focuses on primary and secondary school students. This study extended the research subjects to undergraduates in higher education for the first time. This shift not only demonstrates the continuity and developmental nature of educational research, but also reveals potential differences in digital learning competence across educational stages. By constructing a five-dimensional framework, this study has deepened the understanding of the essential characteristics of digital learning competence. It has clarified the interaction mechanisms among awareness, technology, behavior, management, and evaluation dimensions, providing a solid theoretical foundation for subsequent research.

Second, the difference analysis results for gender and institutional level provide new research perspectives. This study found that male students scored higher than female students in scholarly achievement, practical achievement, and academic achievement. [29] ([29]) proposed that differences in agreeableness, conscientiousness, and negative emotions affect individual digital maturity development. Compared to male students, female students experience more negative emotional arousal ([35]) and consider themselves to have lower general computer self-efficacy ([58]). They show lower enthusiasm for using digital technologies (VR) in teaching and assessment ([38]). Traditional social stereotypes that boys are more suited to learning in science and technology courses also adversely affect female participation enthusiasm in STEM ([43]). In the future, we need more research exploring how to maintain and enhance female students’ interest in digital learning, promote female students’ academic and practical participation in artificial intelligence projects, and improve their digital learning competence and academic achievement.

Meanwhile, in the institutional difference dimension, students from “Double First-Class” universities generally reached higher academic achievement than students from non-Double First-Class institutions. This indicates that digital learning competence development not only depends on individual literacy, but is also deeply influenced by external conditions such as institutional resource support, technological environments, and organizational culture. This reflects the important role of institutional-level digital maturity in student development. [1] ([1]) proposed that university digital maturity includes seven dimensions: leadership, planning and management, quality assurance, scientific research, digital teaching and learning, community service, equipment and technological infrastructure, and technological culture. Compared to non-Double First-Class institutions, “Double First-Class” universities typically possess more advanced digital learning platforms, richer online resource libraries, policies and cultures that encourage teachers to apply digital technologies, and more comprehensive digital skills training support. These factors collectively create more favorable institutional environments for students’ digital learning competence development and academic achievement improvement. This perspective provides important insights for understanding the roles that universities at different levels play in enhancing students’ digital learning competence. It also provides theoretical foundations for future research to further explore associations between institutional support and learning effectiveness.

Additionally, it validates the positive correlation between digital learning competence and academic achievement. This study finds a significant positive correlation between digital learning competence and student academic achievement. Particularly, digital learning evaluation competence has a significant positive predictive effect on academic achievement. This finding is consistent with that rom [63]’s ([63]) research that critical production in undergraduate media literacy has the most significant impact on learning outcomes. This finding not only validates core concepts of Self-Regulated Learning theory—the central role of critical resource assessment and metacognitive reflection in academic success—but also provides important theoretical evidence for educational practitioners. Research indicates that enhancing students’ digital learning competence, especially evaluation capability, is an effective approach to improving academic achievement. Finally, it reveals the importance of critical resource assessment and metacognitive reflection. This study emphasizes the central position of critical resource assessment and metacognitive reflection in digital learning competence. In the era of the information explosion, how students effectively screen, evaluate, and utilize digital resources, and how they optimize learning processes through metacognitive reflection, become key factors determining learning effectiveness. This finding enriches the understanding of digital learning competence and provides theoretical support for cultivating new-era talents with information literacy and autonomous learning abilities. Particularly, it verifies that digital learning evaluation competence has a significant positive predictive effect on academic achievement, aligning with core concepts of Self-Regulated Learning theory and highlighting the central role of critical resource assessment and metacognitive reflection in academic success ([2]; [69]; [22]), helping students escape traps of technology dependence.

Notably, although digital learning awareness and digital learning engagement behaviors are important components of digital learning competence, these two dimensions did not show significant correlations with academic achievement. In this survey, undergraduates scored relatively high in digital learning awareness (M = 5.80), indicating that undergraduates generally recognize the value of digital learning. This may be related to the normalization of online learning after the COVID-19 pandemic. When group awareness levels are generally high, their differential impact on academic achievement may be diluted, resulting in non-significant correlations. According to Self-Regulated Learning Theory ([80]), while “motivation” at the awareness level may stimulate learners’ initial participation willingness, without subsequent technical support, strategic planning, and metacognitive control, this “awareness” often fails to continuously transform into high-quality learning behaviors and outcomes, leading to non-significant correlations between digital learning awareness and academic achievement. Digital learning engagement behaviors such as online communication, resource sharing, or platform usage frequency can reflect students’ participation status in digital environments. However, frequent online participation does not equal deep cognitive processing. According to Cognitive Load Theory (CLT) ([60]), ineffective behavioral participation may increase extraneous cognitive load. Students with high behavioral competence scores in this study may have spent energy on shallow online interactions while neglecting deep knowledge processing, resulting in non-significant correlations between digital learning engagement behaviors and academic achievement. In the context of global higher education digital transformation, there is a need to enhance undergraduates’ digital learning competence to improve their learning effectiveness and performance, thereby improving talent cultivation quality. Therefore, enhancing students’ digital learning competence, especially digital learning evaluation competence, should become an important goal of university general education. Infrastructure and equipment, resources and support, ICT policies, and teacher training are crucial for higher education institutions seeking to implement digital strategies, meet students’ growing digital learning requirements, and maintain competitiveness in the global environment ([12]). Therefore, based on the research findings, this study advocates for the “Education 4.0” concept, calling for integrating cutting-edge technologies and reconstructing educational ecosystems to achieve more personalized, flexible, and lifelong learning experiences. This addresses contradictions between standardized teaching and personalized needs in traditional educational systems, and in adapting to future societal requirements for talent capabilities. Specifically, suggestions for the optimization of university student cultivation processes are proposed in three dimensions: enriching carriers, optimizing methods, and creating environments.

First, in enriching carriers, regarding learners as central roles, paying attention to individual differences, formulating exclusive learning paths through data analysis, and cultivating interdisciplinary and innovative capabilities are required. Universities should integrate digital technology content, such as programming and data analysis, into curriculum systems, enhance students’ technical application abilities through specialized training, and use generative artificial intelligence chatbots and large language models (LLM) to enhance students’ digital learning and boost academic achievement ([57]). Establishing digital learning evaluation courses by systematically integrating capabilities, such as resource assessment, information retrieval, and critical thinking, into professional teaching can strengthen students’ self-evaluation abilities regarding learning processes and outcomes. Moreover, in optimizing methods, we emphasize the combination of personalized and practical teaching. Universities should leverage technologies such as artificial intelligence (AI), big data, virtual reality (VR/AR), blockchain, and Internet of Things (IoT) to optimize teaching processes. Teachers can formulate differentiated teaching plans based on students’ grade levels and ability levels, strengthen basic guidance for lower-grade students, and promote self-regulation ability development in higher-grade students. We can also introduce actual case teaching into classrooms, and promote cross-grade collaborative communication through research competitions, academic salons, and club activities, helping students use digital tools to solve real problems. Last but not least, in creating environments, [25] ([25]) found through an analysis of PISA 2018 data that digital capital accumulation at the school, teacher, and student levels can influence students’ academic achievement. Particularly, ICT software infrastructure—referring to commonly adopted educational platforms and digital tools rather than a specific version—has important positive correlations with student performance. Universities should build intelligent and immersive learning environments, and improve network resources, teaching platforms, and hardware facilities. Constructing supportive cultural ecosystems and providing specialized training and role model inspiration for female students to enhance their digital participation willingness can promote institutional resource sharing to alleviate educational imbalance issues.

The limitations of this study include the following. First, the questionnaire was primarily distributed online, resulting in certain sample selection constraints. Demographic indicators did not include respondents’ socioeconomic backgrounds, geographic distribution, or other information. This may thus not fully represent the digital learning competence and academic achievement situations of all undergraduate student populations. Second, although the study considered control variables such as gender, grade level, and school type, other unidentified or uncontrolled variables may exist. These could include learning motivation, socioeconomic status, family background, etc. Such variables might affect the accuracy of the research results.

Regarding future research prospects, first, with the rapid development of digital technology, the connotations of undergraduate digital learning competence continue to evolve. Future research should deeply analyze the structural connotations of digital learning competence, clarify its core elements, and develop more accurate, comprehensive, and easily operable measurement tools. These would help effectively assess undergraduates’ digital learning competence and provide personalized learning feedback and guidance recommendations. Second, sample diversity and representativeness are crucial for the universality of research results. Future research should seek broader sample populations containing undergraduates from different regions, cultural backgrounds, and educational levels. This would help to explore differences in digital learning competence among different groups and to propose more targeted suggestions to improve undergraduate digital learning competence and academic achievement levels, enhancing the universality of research findings. Third, academic achievement is influenced by multiple factors. To more accurately assess the impact of undergraduate digital learning competence on academic achievement, future research should identify and control as many variables as possible that may affect academic achievement (these include learning motivation, socioeconomic status, family background, etc.) to improve the accuracy of research results. Fourth, this study did not directly measure the digital maturity of higher education institutions. Future research can consider the digital maturity of higher education institutions in the context of digital transformation as an important research direction. Developing and constructing university digital maturity assessment models can be undertaken to help examine the moderating or mediating effects of university digital maturity in the relationship between students’ digital learning competence and academic achievements. Finally, quantitative and qualitative research methods each have their advantages and disadvantages. Future research could attempt to combine both approaches. While collecting extensive data through questionnaire surveys, researchers could also try to obtain more comprehensive research perspectives through interviews, observations, or case studies. This would provide richer insights into and suggestions for educational practice.

## 6. Conclusions

Although digital learning competence has gradually become one of the core essential competencies for students, its specific mechanisms of influence on academic achievement have not been fully explored. This study focused on undergraduate students as research subjects, independently developed and validated measurement tools for digital learning competence and academic achievement with good reliability and validity, collected 312 valid questionnaires, and conducted a systematic empirical analysis.

The results indicate that the surveyed undergraduates demonstrated relatively high overall digital learning competence, with digital learning awareness and behavioral competence being the strongest, while technical competence was relatively weaker. Academic achievement was generally at the upper-middle level, with course achievement and practical achievement outperforming academic achievement. Significant gender differences were observed in academic achievement, practical achievement, and overall academic achievement. Senior students outperformed junior students in terms of course, academic, and comprehensive academic achievements. Students from “Double First-Class” and other key universities demonstrated better academic achievements than those from regular undergraduate institutions. A significant positive correlation exists between digital learning competence and academic achievement. Among these, digital learning evaluation competence as a sub-competence under digital learning competence is particularly prominent, showing significant positive predictive effects on academic achievement. Therefore, this study suggests utilizing multidimensional strategies, including curriculum system optimization, personalized teaching methods, and intelligent learning environment construction, to help university students scientifically apply digital technologies, especially AI technologies, to further enhance undergraduate students’ digital learning capabilities and academic achievement.

## Figures and Tables

**Table 1 behavsci-15-00840-t001:** Dimensions of digital learning competence.

Competence	Component Elements	Description
Digital Learning Competence	Digital Learning Awareness	An individual’s cognition, understanding, and value placed on digital learning, as well as their attitude toward digital learning tools and resources.
	Digital Learning Technical Competence	An individual’s skills and abilities in using digital technologies and tools.
	Digital Learning Engagement Behaviors	A series of behaviors and habits exhibited by individuals during the digital learning process, including collaboration, communication, and connection abilities.
	Digital Learning Self-Management	An individual’s ability to manage learning resources, learning tools, learning time, and learning emotions.
	Digital Learning Evaluation Competence	An individual’s ability to reasonably and accurately evaluate digital learning resources, tools, environments, processes, and outcomes.

**Table 2 behavsci-15-00840-t002:** Descriptive statistical analysis of undergraduate digital learning competence and academic achievement.

Comparative Dimension	Secondary Dimension	N	Minimum	Maximum	Mean	Standard Deviation
Digital Learning Competence	Digital Learning Awareness	312	2.33	7.00	5.80	0.66
	Digital Learning Technical Competence	312	3.33	6.50	5.06	0.76
	Digital Learning Engagement Behaviors	312	3.67	7.00	5.85	0.66
	Digital Learning Self-Management	312	4.00	7.00	5.57	0.65
	Digital Learning Evaluation Competence	312	3.20	7.00	5.59	0.71
	Overall	312	4.25	6.67	5.54	0.47
Academic Achievement	Course Achievement	312	1	14	6.84	2.62
	Scholarly Achievement	312	3	14	5.49	2.39
	Practical Achievement	312	6	30	15.78	4.44
	Overall	312	3.7	14.3	8.22	2.06

**Table 3 behavsci-15-00840-t003:** Differential analysis results of undergraduate academic achievement dimensions.

Comparative Dimension		Course Achievement	Scholarly Achievement	Practical Achievement	Overall Academic Achievement
Gender	Male	7.08 ± 2.75	6.03 ± 2.46	16.44 ± 4.77	8.64 ± 2.01
	Female	6.66 ± 2.51	5.08 ± 2.26	15.28 ± 4.12	7.91 ± 2.05
	F	1.389	3.522	2.297	3.108
	P	0.166	0.000	0.022	0.002
Grade Level	First year	4.54 ± 2.27	3.62 ± 1.20	15.15 ± 4.85	6.38 ± 1.73
	Second year	7.40 ± 2.44	5.87 ± 2.56	15.69 ± 4.13	8.60 ± 2.37
	Third year	7.65 ± 2.56	6.35 ± 2.64	16.35 ± 4.90	9.00 ± 2.08
	Fourth year	6.80 ± 2.56	5.30 ± 2.13	15.63 ± 4.36	8.12 ± 1.82
	Fifth year	7.00 ± 2.62	7.90 ± 3.28	17.80 ± 3.71	9.43 ± 2.52
	F	7.218	9.644	0.905	9.073
	P	0.000	0.000	0.461	0.000
Discipline	Philosophy	7.20 ± 2.59	8.00 ± 2.00	20.40 ± 3.78	10.08 ± 2.42
	Economics	6.94 ± 2.21	5.78 ± 2.69	16.39 ± 4.70	8.48 ± 2.43
	Law	7.00 ± 2.70	5.75 ± 2.60	15.08 ± 3.87	8.24 ± 2.31
	Education	6.60 ± 3.01	5.56 ± 2.20	17.31 ± 5.13	8.43 ± 1.80
	Literature	6.61 ± 2.38	4.89 ± 2.23	15.14 ± 4.70	7.80 ± 1.91
	History	8.20 ± 1.64	5.20 ± 1.30	15.20 ± 3.11	8.70 ± 0.84
	Science	7.00 ± 3.08	5.85 ± 2.46	15.28 ± 3.98	8.31 ± 2.46
	Engineering	7.57 ± 2.77	5.13 ± 2.31	15.63 ± 4.78	8.45 ± 2.12
	Agriculture	9.67 ± 0.58	6.00 ± 0.00	15.33 ± 2.89	9.70 ± 0.87
	Medicine	6.00 ± 2.18	6.29 ± 3.43	17.29 ± 3.36	8.34 ± 2.37
	Military Science	10.00 ± 0.00	7.50 ± 2.12	13.50 ± 0.71	9.95 ± 0.49
	Management	6.36 ± 2.17	5.33 ± 2.31	14.89 ± 3.84	7.75 ± 1.81
	Arts	6.00 ± 0.00	3.00 ± 0.00	15.00 ± 6.25	6.90 ± 1.25
	F	1.393	1.447	1.602	1.272
	P	0.168	0.144	0.090	0.234
University Tier	Project 985	7.65 ± 3.10	5.51 ± 2.20	16.15 ± 5.36	8.71 ± 1.99
	Project 211	8.52 ± 2.60	6.57 ± 2.88	17.59 ± 4.27	9.75 ± 2.26
	“Double First-Class” (excluding 985 & 211)	6.91 ± 2.12	6.32 ± 2.67	16.02 ± 3.85	8.55 ± 1.86
	Other universities	5.83 ± 1.98	4.89 ± 2.04	14.94 ± 3.89	7.37 ± 1.69
	F	18.355	8.321	4.543	21.689
	P	0.000	0.000	0.004	0.000

**Table 4 behavsci-15-00840-t004:** Correlation analysis results between digital learning competence and academic achievement of undergraduates.

	Digital Learning Awareness	Digital Learning Technical Competence	Digital Learning Engagement Behaviors	Digital Learning Self-Management	Digital Learning Evaluation Competence	Digital Learning Competence	Course Achievement	Academic Achievement	Practical Achievement	Overall Academic Achievement
Digital Learning Awareness	1									
Digital Learning Technical Competence	0.357 **	1								
Digital Learning Engagement Behaviors	0.288 **	0.127 *	1							
Digital Learning Self-Management	0.278 **	0.306 **	0.118 *	1						
Digital Learning Evaluation Competence	0.313 **	0.429 **	0.218 **	0.377 **	1					
Digital Learning Competence	0.711 **	0.760 **	0.425 **	0.593 **	0.726 **	1				
Course Achievement	0.041	0.058	−0.053	0.097	0.138 *	0.095	1			
Academic Achievement	0.01	0.152 **	−0.063	0.157 **	0.166 **	0.143 *	0.310 **	1		
Practical Achievement	0.043	0.218 **	0.081	0.068	0.199 **	0.196 **	0.086	0.354 **	1	
Academic Achievement	0.048	0.184 **	−0.021	0.146 **	0.231 **	0.194 **	0.780 **	0.697 **	0.608 **	1

* *p* < 0.05 ** *p* < 0.01.

**Table 5 behavsci-15-00840-t005:** Regression analysis results of digital learning competence and academic achievement of undergraduates.

Model	Unstandardized Coefficients		t	Sig.	Collinearity Statistics	
	B	Std. Error			Tolerance	VIF
Independent Variable						
(Constant)	4.673	0.840	5.562	0.000		
Digital Learning Evaluation Competence	0.480	0.149	3.215	0.001	0.969	1.032
Control Variables						
University Tier						
Double First-Class Universities	1.458	0.214	6.828	0.000	0.965	1.036
Other Universities	0					
Grade Level						
Lower Grades	−0.638	0.252	−2.533	0.012	0.981	1.019
Higher Grades	0					
Gender						
Male	0.510	0.214	2.385	0.018	0.976	1.024
Female	0					
R^2^			0.208			
Adjusted R^2^			0.198			
F			17.462			
P			0.000			
Dependent Variable: Academic Achievement						

## Data Availability

The raw data supporting the conclusions of this article will be made available by the authors on request.

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
