# Peer review of "The Impact of Digital Learning Competence on the Academic Achievement of Undergraduate Students"

_behavsci, 2025, doi:10.3390/bs15070840_

Round 1
Reviewer 1 Report
Comments and Suggestions for Authors
This manuscript explores a relevant topic for the field of educational psychology by analyzing the relationship between digital learning competence and academic achievement in undergraduate students. The study is grounded on a solid literature review, offers a clear conceptual framework, and employs a valid and reliable instrument. The statistical analysis is appropriate and supports the main findings.
The paper contributes to the literature by distinguishing between different dimensions of digital competence, especially highlighting the predictive role of evaluation competence. This finding reinforces theoretical perspectives on self-regulated learning and offers actionable insights for educational practice.
For improvement, I suggest:
C1: Clarifying the construct of “digital learning competence” in contrast to broader notions like "digital literacy" or "digital maturity".
C2: Providing a deeper reflection on the gender-based and institutional differences found, which currently remain underdeveloped.
C3: Refining language in a few sections to improve fluency and academic tone, although the English is generally clear.
C4: One aspect that could be further addressed is the institutional digital maturity.
C5: It could be valuable if you relate your findings to current advances in the literature on digital transformation and digital maturity.
C6: I highly suggest linking how this "universities can improve existing student devel-23 opment processes through measures such as enriching carriers, optimizing methods, and 24 creating supportive environments to foster undergraduates’ digital learning competence, 25 thereby enhancing their academic achievement." is linked to the concept of Education 4.0 or Education 5.0.
Overall, the paper presents meaningful results and practical implications for higher education. I recommend acceptance after minor revisions.
Author Response
Thank you very much for taking the time to review this manuscript. Your valuable and insightful comments have been of great help to us in improving the quality of this paper. We fully accept all the suggested revisions and have carefully incorporated them into the manuscript. The newly revised text has added 3,552 words and 20 new references. The modified parts have been highlighted in red for your easy reference. Please review the revised version for the detailed changes. Once again, we sincerely appreciate your constructive feedback and guidance.
Comments 1: Clarifying the construct of “digital learning competence” in contrast to broader notions like "digital literacy" or "digital maturity".
Response 1: Thank you for pointing this out. We agree with this comment. Therefore, we have added a dedicated section in the revised manuscript to clarify the construct of “digital learning competence” and distinguish it from broader concepts such as "digital literacy" and "digital maturity". This new section can be found on page 3, starting from paragraph 1, line 108-131,The revised parts are marked in red in the modified manuscript. Additionally, we have included the specific items of each sub-scale of the Digital Learning Competence measurement tool in the appendix, thereby making its constituent elements more explicit and tangible.
Comments 2: Providing a deeper reflection on the gender-based and institutional differences found, which currently remain underdeveloped.
Response 2: Thank you for your insightful comment. We fully concur with the need to delve deeper into the gender-based and institutional differences. Consequently, we have significantly expanded the analysis of these differences throughout the manuscript.
In the 5. Discussion section, we have added a dedicated paragraph to conduct a more in - depth reflection on the results of gender differences and school - level differences found in the study. By integrating existing literature, we explored the factors contributing to these differences and proposed research perspectives and practical strategies for addressing these differences in the future. This new section can be found on page 18-19, starting from paragraph 3, line 46-79, The revised parts are marked in red in the modified manuscript.
Comments 3: Refining language in a few sections to improve fluency and academic tone, although the English is generally clear.
Response 3: Thank you for your valuable feedback. We wholeheartedly agree that refining the language can further enhance the quality of the manuscript. In response to your comment, we have meticulously reviewed and revised several sections to improve fluency and strengthen the academic tone.
Comments 4: One aspect that could be further addressed is the institutional digital maturity.
Response 4: Thank you for your constructive comment. We fully recognize the importance of delving deeper into institutional digital maturity, and we appreciate your suggestion. In response, we have made several substantial revisions to expand and enrich the discussion on this aspect.
In the 2. Literature Review section and the newly added paragraphs reflecting on school - level differences in the 5. Discussion section, we have integrated the concept of "digital maturity of higher education institutions." In the practical strategies part of the Discussion section, we have put forward relevant suggestions on how higher education institutions can enhance their digital maturity in the context of digital transformation to promote the development of students' digital learning competence and academic achievements. In the section on future research prospects in the Discussion section, we have also clearly pointed out that future research can take the digital maturity of higher education institutions as an important research direction. This new section can be found on page 3, starting from paragraph 1, line 109-129; page 19, starting from paragraph 1, line 61-78. The revised parts are marked in red in the modified manuscript.
Comments 5: It could be valuable if you relate your findings to current advances in the literature on digital transformation and digital maturity.
Response 5: Thank you for your insightful and valuable comment. We wholeheartedly agree that linking our findings to the current advances in the literature on digital transformation and digital maturity can significantly enhance the relevance and impact of our research. In response to this suggestion, we have made the following comprehensive revisions. In the first part and the literature review section, we have added an overview of the latest research progress in the fields of digital transformation and digital maturity. Meanwhile, in the section of countermeasures and suggestions for enhancing digital learning competence within the discussion part, we have engaged in dialogues with relevant up - to - date literature. This new section can be found on page 3, starting from paragraph 1, line 108-131; page 19, starting from paragraph 2, line 61-86. The revised parts are marked in red in the modified manuscript.
Comments 6: I highly suggest linking how this "universities can improve existing student devel-23 opment processes through measures such as enriching carriers, optimizing methods, and 24 creating supportive environments to foster undergraduates’ digital learning competence, 25 thereby enhancing their academic achievement." is linked to the concept of Education 4.0 or Education 5.0.
Response 6: Thank you for your highly valuable suggestion. We wholeheartedly agree that establishing a clear link between our proposed measures for universities and the concepts of Education 4.0 can greatly deepen the theoretical significance and practical implications of our research. In response to this comment, we have made the following targeted revisions: In the section of countermeasures and suggestions, we explicitly propose “Education 4.0” concept, calling for integrating cutting-edge technologies and reconstructing educational ecosystems to achieve more personalized, flexible, and lifelong learning experiences. This addresses contradictions between standardized teaching and personalized needs in traditional educational systems, adapting to future societal requirements for talent capabilities.” and put forward specific countermeasures and suggestions in combination with "Education 4.0". This new section can be found on page 20, starting from paragraph 1, line 132-166. The revised parts are marked in red in the modified manuscript.

Reviewer 2 Report
Comments and Suggestions for Authors
The article presents the study’s objectives and methodology clearly; however, there are several key issues that need to be addressed before it can be considered for publication:
-
Lack of clarity regarding Digital Learning Competence: While the authors refer to a framework involving Digital Learning Competence and its components, they do not specify the descriptors that constitute these elements. Furthermore, there is no explanation of how these components were assessed. Providing clear definitions and operationalizations is essential for understanding the study's foundation and replicability.
-
Insufficient detail on ‘academic achievement’: The same concern applies to the concept of academic achievement. The authors mention three aspects that comprise it, yet these are not described in sufficient detail. The methodology lacks transparency in how academic achievement was defined, measured, and analyzed. Without this information, it is difficult to assess the validity and significance of the findings.
-
Unaddressed research question: The third research question remains unanswered in the results section. Although it is touched upon in the discussion, the lack of a concrete proposal or analytical response undermines the completeness of the study. Additionally, references to the role of institutions are made without citing any supporting literature. I recommend consulting and citing relevant work in this area, such as the following paper:
- Esteve-Mon, Francesc; Postigo-Fuentes, Ana Yara & Castañeda, Linda (2022) A strategic approach of the crucial elements for the implementation of digital tools and processes in Higher Education. Higher Education Quarterly. https://doi.org/10.1111/hequ.12411
Addressing these issues will substantially strengthen the study's methodological rigor and overall contribution.
Author Response
Thank you very much for taking the time to review this manuscript. Your valuable and insightful comments have been of great help to us in improving the quality of this paper. We fully accept all the suggested revisions and have carefully incorporated them into the manuscript. The newly revised text has added 3,552 words and 20 new references. The modified parts have been highlighted in red for your easy reference. Please review the revised version for the detailed changes. Once again, we sincerely appreciate your constructive feedback and guidance.
Comments 1: Lack of clarity regarding Digital Learning Competence: While the authors refer to a framework involving Digital Learning Competence and its components, they do not specify the descriptors that constitute these elements. Furthermore, there is no explanation of how these components were assessed. Providing clear definitions and operationalizations is essential for understanding the study's foundation and replicability.
Response 1: Thank you for your meticulous review and this crucial comment. We fully recognize the significance of clearly defining and operationalizing Digital Learning Competence, and we deeply appreciate your pointing out these issues. To address your concerns, we have made the following comprehensive revisions: We have clarified the concept of digital learning competence in Section 2.1 "Digital Learning Competence". In the appendix, a complete measurement scale for digital learning competence has been added, with detailed descriptions of the specific items for the five dimensions of digital learning competence. In Section 3.3 "Collection of Formal Data", the assessment method of the digital learning competence scale has been explained. This new section can be found on page 3, starting from paragraph 1, line 108-131; page 9, starting from paragraph 3, line 361-395; Appendix 1. The revised parts are marked in red in the modified manuscript.
Comments 2: Insufficient detail on ‘academic achievement’: The same concern applies to the concept of academic achievement. The authors mention three aspects that comprise it, yet these are not described in sufficient detail. The methodology lacks transparency in how academic achievement was defined, measured, and analyzed. Without this information, it is difficult to assess the validity and significance of the findings.
Response 2: Thank you for your insightful and valuable comment. We wholeheartedly acknowledge the importance of providing a comprehensive and detailed account of the concept of academic achievement, and we sincerely appreciate your highlighting these areas for improvement. In response to your concerns, we have made the following extensive revisions: We have provided detailed descriptions of the definitions of curriculum achievement, academic achievement, and practical achievement in the section "3.2.2 Development of the Academic Achievement Questionnaire". In the section "3.3 Collection of Formal Data", we have given a detailed explanation of the measurement and analysis of academic achievement. Additionally, we have included a detailed strategic plan regarding this concept in the appendix. This new section can be found on page 9, starting from paragraph 2, line 340-359; page 9, starting from paragraph 3, line 361-395; Appendix 1. The revised parts are marked in red in the modified manuscript.
Comments 3: Unaddressed research question: The third research question remains unanswered in the results section. Although it is touched upon in the discussion, the lack of a concrete proposal or analytical response undermines the completeness of the study. Additionally, references to the role of institutions are made without citing any supporting literature. I recommend consulting and citing relevant work in this area, such as the following paper: Esteve-Mon, Francesc; Postigo-Fuentes, Ana Yara & Castañeda, Linda (2022) A strategic approach of the crucial elements for the implementation of digital tools and processes in Higher Education. Higher Education Quarterly. https://doi.org/10.1111/hequ.12411
Response 3: We sincerely appreciate your meticulous review and these constructive comments. We fully recognize the issues you have pointed out regarding the unanswered research question and insufficient literature citation, and we have taken immediate steps to address them comprehensively. We carefully read the article A strategic approach of the crucial elements for the implementation of digital tools and processes in Higher Education. In the section of countermeasures and suggestions within the discussion part, we further emphasized and responded to the third research question of this study, "How can we improve undergraduates’ digital learning competence to enhance their learning outcomes and academic performance, thereby improving talent cultivation quality?" Additionally, we added citations of literature when analyzing the role of institutions. This new section can be found on page 19, starting from paragraph 2, line 61-79; page 20, starting from paragraph 1, line 104-166. The revised parts are marked in red in the modified manuscript.

Reviewer 3 Report
Comments and Suggestions for Authors
The manuscript investigates the relationship between digital learning competence and academic achievement among undergraduate students. This is a timely and relevant topic, especially in light of the ongoing digital transformation in higher education. The manuscript presents a clear research design and a well-structured measurement development process, contributing to a growing body of knowledge in digital education research.
The study demonstrates potential for publication, but in its current form, it would benefit from a number of revisions to enhance its theoretical depth, analytical clarity, and alignment between results and conclusions. My comments below are intended to support the refinement of your work and to help strengthen its impact and academic rigor.
Comment 1:
The literature review presents several frameworks and policy references (e.g., DigComp, OECD, ISTE), but lacks critical integration. For instance, the section on page 2–3 describes various competence models without clearly justifying why the proposed five-dimensional model was chosen over others. Moreover, concepts such as self-regulation or digital fluency could be better linked to the structure of the instrument.
Recommendation: Provide a clearer rationale for the dimensions included in the study, and consider positioning your framework in dialogue with previous empirical models, especially those focused on higher education students.
Comment 2 :
The manuscript concludes that digital learning competence positively impacts academic achievement in general. However, the regression model (p. 14) shows that only the evaluation competence dimension is a statistically significant predictor. Other dimensions were excluded from the model due to lack of predictive power.
Recommendation: Moderate the claims in the abstract, conclusion, and discussion. Clearly state that the effect found pertains specifically to the evaluation dimension, and avoid generalizations not supported by the data.
Comment 3 :
While the manuscript includes many citations, several are descriptive or policy-related. The engagement with recent empirical studies (especially from 2023–2024) that explore digital learning and academic performance is limited.
Recommendation: Enrich the literature review and discussion by incorporating more recent international research that offers theoretical and empirical depth on the constructs analyzed.
Comment 4 :
The discussion mainly reiterates statistically significant results but does not reflect on the non-significant ones. For example, the absence of correlation between digital awareness or engagement behaviors and academic achievement is not explored.
Recommendation: Include possible interpretations of these null findings. This would enhance the explanatory value of the study and demonstrate theoretical reflexivity.
Comment 5:
While the development of the questionnaire is well documented, the manuscript could be improved by including examples of items from each dimension, or at least summarizing item types to illustrate what was measured.
Recommendation: Consider adding example items from each subscale of the digital learning competence instrument, either in the text or as an appendix.
Final comment:
The manuscript presents original and relevant research with a solid methodological base. With deeper theoretical contextualization, more balanced discussion of findings, and a review of recent literature, the paper could make a strong contribution. I encourage the authors to revise carefully in light of these comments.
Author Response
Thank you very much for taking the time to review this manuscript. Your valuable and insightful comments have been of great help to us in improving the quality of this paper. We fully accept all the suggested revisions and have carefully incorporated them into the manuscript. The newly revised text has added 3,552 words and 20 new references. The modified parts have been highlighted in red for your easy reference. Please review the revised version for the detailed changes. Once again, we sincerely appreciate your constructive feedback and guidance.
Comments 1: The literature review presents several frameworks and policy references (e.g., DigComp, OECD, ISTE), but lacks critical integration. For instance, the section on page 2–3 describes various competence models without clearly justifying why the proposed five-dimensional model was chosen over others. Moreover, concepts such as self-regulation or digital fluency could be better linked to the structure of the instrument.
Recommendation: Provide a clearer rationale for the dimensions included in the study, and consider positioning your framework in dialogue with previous empirical models, especially those focused on higher education students.
Response 1: We are truly grateful for your insightful review and this valuable feedback. We fully acknowledge the issues you’ve raised regarding the lack of critical integration in the literature review, and we’ve made targeted and comprehensive revisions to address these concerns. We have critically integrated existing digital learning competence models in both the literature review section and the section "3.2.1 Development of the Digital Learning Competence Scale". By combining the self - regulation theory, we have elucidated the appropriateness of adopting the five - dimensional model in this study. This new section can be found on page 3, starting from paragraph 1, line 108-131 ; page 4, starting from paragraph 2, line 244-304,The revised parts are marked in red in the modified manuscript.
Comments 2: The manuscript concludes that digital learning competence positively impacts academic achievement in general. However, the regression model (p. 14) shows that only the evaluation competence dimension is a statistically significant predictor. Other dimensions were excluded from the model due to lack of predictive power.
Recommendation: Moderate the claims in the abstract, conclusion, and discussion. Clearly state that the effect found pertains specifically to the evaluation dimension, and avoid generalizations not supported by the data.
Response 2: We sincerely appreciate your astute observations and this constructive feedback. We are fully aware of the problem of overgeneralization between our previous broad conclusions and the specific results of the regression model, and we have made comprehensive and targeted revisions to address this issue. We have rephrased the content in the abstract, conclusion, and discussion sections, clearly stating that the sub - competence of digital learning evaluation within digital learning competence has a significant positive predictive effect on the academic achievements of undergraduates. When other factors remain constant, for every one-unit increase in digital learning evaluation ability, academic achievement increases by 0.480 units. This new section can be found on page 1, starting from paragraph 1, line 19-23 ; page 22, starting from paragraph 2, line 220-223,The revised parts are marked in red in the modified manuscript.
Comments 3: While the manuscript includes many citations, several are descriptive or policy-related. The engagement with recent empirical studies (especially from 2023–2024) that explore digital learning and academic performance is limited.
Recommendation: Enrich the literature review and discussion by incorporating more recent international research that offers theoretical and empirical depth on the constructs analyzed.
Response 3: We are extremely grateful for your perceptive review and this valuable suggestion. We fully acknowledge the limitation in our manuscript regarding the insufficient engagement with recent empirical studies on digital learning and academic performance, and we have taken immediate and comprehensive actions to address this issue. We have added the latest research findings on digital learning and academic performance from 2023 - 2024 in both the literature review and discussion sections. A total of 20 new references have been incorporated, further enhancing the richness of the literature. This new section can be found on page 3, starting from paragraph 1, line 108-131 ; page 6, starting from paragraph 2, line 244-304; page 22, starting from paragraph 2, line 220-223; The revised parts are marked in red in the modified manuscript.
Comments 4: The discussion mainly reiterates statistically significant results but does not reflect on the non-significant ones. For example, the absence of correlation between digital awareness or engagement behaviors and academic achievement is not explored.
Recommendation: Include possible interpretations of these null findings. This would enhance the explanatory value of the study and demonstrate theoretical reflexivity.
Response 4: We sincerely appreciate your incisive review and this crucial suggestion. We fully recognize the oversight in our original discussion section regarding the lack of analysis of non - significant results, and we have made substantial and targeted revisions to address this issue. We have added reflections on the non-significant results, clearly stating that "although digital learning awareness and behavioral competence are important components of digital learning competence, there is no significant correlation between these two dimensions and academic achievement. In this survey, undergraduate students scored relatively high on the digital learning awareness competence dimension (M=5.80), indicating that undergraduate students generally recognize the value of digital learning. This may be related to the normalization of online learning after the COVID-19 pandemic. When the awareness level of the group is generally high, its differential impact on academic achievement may be diluted, resulting in a non-significant correlation. According to the self-regulated learning theory (Zimmerman, 2000), although the'motivation' at the awareness level may stimulate learners' initial willingness to participate, without subsequent technical support, strategic planning, and metacognitive control, this 'awareness' is often difficult to be continuously transformed into high-quality learning behaviors and outcomes, leading to the non-significant correlation between digital learning awareness competence and academic achievement. Digital learning behavioral competence, such as the frequency of online communication, resource sharing, or platform use, can reflect students' participation status in the digital environment. However, frequent online participation does not necessarily mean deep cognitive processing. According to the Cognitive Load Theory (CLT) (Sweller et al., 2011), ineffective behavioral participation may increase extraneous cognitive load. Students with high scores in behavioral competence in this study may have spent their energy on superficial online interactions, but ignored the deep processing of knowledge, resulting in the non-significant correlation between digital learning behavioral competence and academic achievement." This new section can be found on page 20, starting from paragraph 1, line 104-139. The revised parts are marked in red in the modified manuscript.
Comments 5: While the development of the questionnaire is well documented, the manuscript could be improved by including examples of items from each dimension, or at least summarizing item types to illustrate what was measured.
Recommendation: Consider adding example items from each subscale of the digital learning competence instrument, either in the text or as an appendix.
Response 5: We are truly grateful for your meticulous review and this insightful suggestion. We fully acknowledge that providing concrete examples of questionnaire items can significantly enhance the clarity of our measurement methods, and we have taken immediate steps to incorporate this valuable feedback. At the end of the paper, we added Appendix 1 to provide a detailed description of the measurement plan for the dimensions of the core variables in the paper. This new section can be found in Appendix 1. The revised parts are marked in red in the modified manuscript.

Round 2
Reviewer 2 Report
Comments and Suggestions for Authors
-
Reviewer 3 Report
Comments and Suggestions for Authors
We appreciate the revisions made to the manuscript. All the suggested changes have been carefully addressed, and we find the modifications to be appropriate and well implemented. The updated version presents clearer arguments, improved structure, and a more precise use of terminology. We are satisfied with the current version